# Biocidal Activity of a Nanoemulsion Containing Essential Oil from *Protium heptaphyllum* Resin against *Aedes aegypti* (Diptera: Culicidae)

**DOI:** 10.3390/molecules26216439

**Published:** 2021-10-25

**Authors:** Cleidjane Gomes Faustino, Fernando Antônio de Medeiros, Allan Kardec Ribeiro Galardo, Alex Bruno Lobato Rodrigues, Anderson Luiz Pena da Costa, Rosany Lopes Martins, Lethicia Barreto Brandão, Lizandra Lima Santos, Marcos Antônio Alves de Medeiros, Patrick de Castro Cantuária, Ana Luzia Ferreira Farias, Jader Santos Cruz, Sheylla Susan Moreira da Silva de Almeida

**Affiliations:** 1Postgraduate Program of Pharmaceutical Innovation, Laboratory of Pharmacognosy and Phytochemistry, Federal University of Amapá, Highway Juscelino Kubitschek, Macapá 68902-280, Brazil; cgfenfermagem@gmail.com (C.G.F.); pena.pharmacist91@gmail.com (A.L.P.d.C.); lethiciabrandao12@gmail.com (L.B.B.); lizandralsantos@gmail.com (L.L.S.); analuziafarias@yahoo.com.br (A.L.F.F.); 2Postgraduate Program in Health Sciences, Federal University of Amapá, Highway Juscelino Kubitschek, Macapá 68902-280, Brazil; fernandomedeiros1973@gmail.com; 3Institute of Scientific and Technological Research of the State of Amapá (IEPA), Highway Juscelino Kubitschek, Macapá 68903-419, Brazil; allangalardo@gmail.com (A.K.R.G.); patrickcantuaria@gmail.com (P.d.C.C.); 4Postgraduate Program in Biodiversity and Biotechnology Network BIONORTE, Laboratory of Pharmacognosy and Phytochemistry, Federal University of Amapá, Highway Juscelino Kubitschek, Macapá 68902-280, Brazil; alexrodrigues.quim@gmail.com (A.B.L.R.); rosyufpa@gmail.com (R.L.M.); 5Faculty of Medical Sciences of Paraíba BR 230, Km 9, Intermares-Cabedelo, Paraíba 58310-000, Brazil; marcosmaa@gmail.com; 6Nova Esperança (FAMENE) School of Medicine, Frei Galvão, Gramame, João Pessoa 58067-695, Brazil; 7Department of Biochemistry and Immunology, Federal University of Minas Gerais, Avenida Antônio Carlos, Belo Horizonte 31270-901, Brazil; jadercruzytrio@gmail.com

**Keywords:** *Protium heptaphyllum*, essential oil, *Aedes aegypti*, biocidal activities

## Abstract

This work aimed to prepare a nanoemulsion containing the essential oil of the *Protium heptaphyllum* resin and evaluate its biocidal activities against the different stages of development of the *Aedes aegypti* mosquito. Ovicide, pupicide, adulticide and repellency assays were performed. The main constituents were *p*-cymene (27.70%) and *α*-pinene (22.31%). The developed nanoemulsion showed kinetic stability and monomodal distribution at a hydrophilic–lipophilic balance of 14 with a droplet size of 115.56 ± 1.68 nn and a zeta potential of −29.63 ± 3.46 mV. The nanoemulsion showed insecticidal action with LC_50_ 0.404 µg·mL^−1^ for the ovicidal effect. In the pupicidal test, at the concentration of 160 µg·mL^−1^, 100% mortality was reached after 24 h. For adulticidal activity, a diagnostic concentration of 200 µg·mL^−1^ (120 min) was determined. In the repellency test, a concentration of 200 µg·mL^−1^ during the 180 min of the test showed a protection index of 77.67%. In conclusion, the nanobiotechnological product derived from the essential oil of *P. heptaphyllum* resin can be considered as a promising colloid that can be used to control infectious disease vectors through a wide range of possible modes of applications, probably as this bioactive delivery system may allow the optimal effect of the *P. heptaphyllum* terpenes in aqueous media and may also induce satisfactory delivery to air interfaces.

## 1. Introduction

*Aedes aegypti*, belonging to the Diptera order and the Culicidae family, is considered to be one of the vectors responsible for the transmission of arboviruses such as Dengue, Chikungunya and Zika, which have high mortality rates. The spread of *A. aegypti* is associated with the disorderly growth of cities and the lack of basic sanitation and housing, favoring mosquito breeding sites [1,2].

Controlling *A. aegypti* has been a major and important challenge, especially in emerging countries. Even in situations where resources have been specifically released for the application of programs to combat the vector, most of the time they have not been successful. Strategies for the chemical control of mosquitoes in the egg, larva and adult phases are promising through the presence of phytochemicals present in essential oils, which is an alternative mechanism of action to commercial insecticides [3]. *P. heptaphyllum* belongs to the Burseraceae family and it has a wide use in popular medicine. Its essential oil has biocidal potential against *A. aegypti* at different stages of life [4].

Currently, the growing concern within society for products that are increasingly safe for the environment and humans, therefore causing lower harmful effects to non-target organisms, has led to a resurgence in the search for plant-derived insecticides [5]. Among the natural products for insecticidal purposes, essential oils are considered important candidates for the control of arthropods of medical importance [6].

Essential oils from medicinal and aromatic plants are protected against pathologies and contain antioxidants. In research carried out with Extra Virgin Olive Oil, the biological properties of (EVOO) were unchanged in the long-term period; it was then decided to investigate the effect of adding medicinal plants to prevent oxidative processes. Improving the shelf life of foods and the aromatic flavor of EVOO involved the addition of different essential oils such as Sicilian sage, oregano, rosemary and thyme. Offering good, healthy and safe products with a balanced nutritional profile that are economically accessible to all consumers is important [7].

However, essential oils present a challenge for practical application due to the need for high concentrations to obtain the desired effect and their chemical instability if applied directly. Therefore, to overcome these challenges, essential oil-based aqueous nanoemulsions are used as delivery systems for bioactive molecules composed of an oily phase that contains a surfactant and a bioactive lipophilic compound, in this case, essential oil (OE) in an aqueous phase. Nanoemulsions are kinetically stable for a longer period. The small size of the particle makes it difficult to deform, further reducing coalescence instability [8].

Most nanoproducts have a series of advantages, such as greater stability, favorable organoleptic characteristics, membrane permeability, bioavailability, increased solubility in the water of less soluble substances and even the controlled release of substances. In addition to that, nanoemulsions are a type of formulation in the group of emulsions that stand out as they are relatively easy to obtain by different methods [9].

Thus, the objectives of this study are to evaluate biocidal activities including ovicide, pupicide, adulticide and repellency against the *A. aegypti* mosquito [10].

## 2. Results

### 2.1. Chemical Composition of the Essential Oil Obtained from the P. heptaphyllumresin

The yield of the *P. heptaphyllum* resin essential oil was 0.69 ± 0.08 (*m/m*) with the transparent color and a strong aroma characteristic of the species. The phytochemical profile revealed the presence of 20 compounds. Among them, *p*-cymene (27.70%) and *α*-pinene (22.31%) were more abundant (Figure 1).

### 2.2. Nanoemulsion Stability of the P. heptaphyllum Resin Essential Oil

The optimized nanoemulsion prepared with the essential oil of the *P. heptaphyllum* resin by the low-energy, non-heating, organic solvent-free method presented a bluish reflect. No sedimentation and no phase separation were observed and therefore, the overall macroscopic visual characteristics of potential kinetic stability nanoemulsions were revealed.

The potential kinetic stability at a hydrophilic–lipophilic balance (HLB) of 14 was also suggested by photon correlation spectroscopy and revealed that the *P. heptaphyllum* essential oil nanoemulsion (PHEON) had a droplet size of 109 ± 0.75 nm, a polydispersity index of 0.29 ± 0.007 and a zeta potential of −21.7 ± 1.10 mV immediately after production.

After 14 days, the particle size was 115.56 ± 1.68 nm, the polydispersity index was equal to 0.40 ± 0.005 and the zeta potential was −29.63 ± 3.46 mV. It should be highlighted that the enhancement of the droplet size after 14 days, when compared to day 0, corresponded to only 5.34% and the slight PdI augmentation may be related to factors such as the main profile of the droplet distribution across a time period by the potential suitable kinetic stability of the nanoemulsion for this study.

### 2.3. Ovicidal Activity Assay

The results of the hatching rate of the *A. aegypti* eggs showed that PHEON had ovicidal potential, since with the highest concentrations evaluated (20 and 15 µg·mL^−1^) there was no hatching of eggs in 24 and 48 h, while in 10.5 and 2 µg·mL^−1^, the hatching rate was below 15% both in 24 and 48 h of exposure.

The hatching percentage was reduced and equal to LC_50_ 0.404 µg·mL^−1^ in 24 and LC_50_ 0.482 µg·mL^−1^ in 48 h (concentration at which 50% of the *A. aegypti* eggs showed a lethal effect in a defined time), resulting in a high value when compared to the standard larvicidal Esbiothrin, with an LC50 of 0.0034 µg·mL^−1^. This result demonstrates the biocidal potential of PHEON in the first development phase of *A. aegypti* (Table 1).

### 2.4. Pupicidal Activity Assay

Table 2 shows the CL_50_ and CL_90_ values of *A. aegypti* pupae in contact with PHEON at 24 h and 48 h of exposure. The concentrations of 80 and 90 µg·mL^−1^ had a mortality rate below 50% and 100 µg·mL^−1^ demonstrated 100% mortality in a 48 h assay. Concentrations of 130 and 160 µg·mL^−1^ reached maximum mortality in 24 h.

### 2.5. Adulticidal Activity Assay

The results indicate that *A. aegypti* mosquitoes were susceptible to 100% mortality at a concentration of 200 µg·mL^−1^ in 120 min of exposure to the product. In the same exposure period, concentrations of 100 and 150 µg·mL^−1^ showed a mortality of approximately 46% and 70%. Thus, it can be concluded that the nanoemulsion had a diagnostic concentration of 200 µg·mL^−1^ in a diagnostic time of 120 min, as shown in Figure 2.

### 2.6. Repellent Activity Assay

The Figure 3 below shows a comparison of the Protection Index of different concentrations of the tested nanoemulsion and the controls depending on the application time. During the test, the positive control did not present any sting and/or landing. In 30 min after the application of the product in the different concentrations, a protection index of 89.34%, 97.67% and 98.67% was demonstrated, whereas in the time of 60, 90, 120, 150 and 180 min, the results in the three concentrations 100, 150 and 200 µg·mL^−1^ were significantly smaller compared to the control. All tested concentrations at the 180 min application protected around 73.34%, 75.00% and 77.67%.

## 3. Discussion

### 3.1. The Chemical Composition and Nanoemulsion Stability of the Essential Oil

The species of the Burseraceae family are commercially attractive, can be sustainably exploited in the Amazon region through forest management and are widely exploited economically as a source of raw material for several areas of the industry due to the production of resin containing essential oils and triterpenes [12]. Essential oils are volatile and contain several chemical compounds, which individually or together have insecticidal activity. The chemical composition of the essential oil of the *P. heptaphyllum* resin presented *p-cymene* (27.70%) and *α*-pinene (22.31%) as major constituents, but in this analysis, all minor and major constituents were identified [10], which corroborates with the data found in the literature concerning the specific chemical composition [13].

Nanoemulsions are kinetically stable colloidal systems of two immiscible liquids stabilized by surfactants. Thus, the nanoemulsion loaded with the *P. heptaphyllum* resin essential oil presented no obvious signs of microscopic or macroscopic instability in the evaluated time [10]. Several methods have been used to combat *A. aegypti*; the application of chemical substances to control its life cycle is one of them. In this context, the Burseraceae family is one of the alternatives for the possible control of vectors; it is, therefore, extremely important for public health. The discovery of new substances that are effective in pest prevention and offer safety and economic viability is rather important especially because they would have a low environmental impact [14].

### 3.2. Ovicidal Evaluation of the P. heptaphyllum Nanoemulsion in A. aegypti

*A. aegypti* eggs have an elongated shape and measure about 0.4 mm. At the time of laying the eggs are white but they quickly darken and become black. The stiffness of the chorion increases when they become older and this is responsible for their resistance to dehydration, extreme temperatures and ultraviolet radiation and their tolerance to the action of pathogens and products with ovicidal activity, contributing to the effective control of the mosquito [15].

Some external factors can interfere with ovicidal activity such as the age of the egg, the concentration and the period of exposure to the products used in the control [16]. A study carried out with *Crotalaria pallida* inhibited the hatching of eggs by 95% through the methanolic extract, by 90% in the ethanolic extract by contact and by 100% in the hatching test at a concentration of 500 µg·mL^−1^ [16]. In another study, the essential oils of *Ricinus communis* and *Cnidoscolus phyllacantus* influenced the embryogenesis of *A. aegypti* eggs and presented hatching rates of 60% for *R.communis* and 95% for *C. phyllacantus* [17].

The ovicidal effect of five essential oils was evaluated at concentrations of 100, 10, 1 and 0.1 µg·mL^−1^ for the species *Mentha piperita*, *Ocimum basilicum*, *Rosemarinus officinalis*, *Cymbopogon nardus* and *Apium graveolens*. All showed better ovicidal activity at higher concentrations [18]. Nanobiotechnological insecticides have also been mentioned in the scientific literature as the nanoemulsion of *Citrus hystrix* oil at a concentration of 1000 µg·mL^−1^ resulted in a percentage of 92.8%, while at 62.5 µg·mL^−1^ it achieved 85.6% [19].

The results presented in this study and contextualized with those available in the scientific literature demonstrate the nanobiotechnological potential of *P. heptaphyllum* nanoemulsion in the control of the development of *A. aegypti* eggs.

### 3.3. Pupicidal Evaluation of P. heptaphyllum Nanoemulsion in A. aegypti

The evaluation of the effect of the essential oil *P. heptaphyllum* nanoemulsion on *A. aegypti* pupae indicated significant results confirmed by the lethal concentration (LC_50_ and LC_90_) obtained after 24 and 48 h of exposure to the nanoemulsion.

The pupicidal potential of *P. heptaphyllum* nanoemulsion can be compared with data reported by others. *A. coriacea* had a mortality rate of 62.5% in *A. aegypti* pupae at 500 µg·mL^−1^ [18] and *Condonopsis javanica* demonstrated 75% mortality of *A. albopictus* pupae at 60,000 µg·mL^−1^ [19]. The effect of the *Citrus hystrix* essential oil nanoemulsion shows that the highest mortality of *A. aegypti* pupae occurred at a concentration of 1000 µg·mL^−1^ and was equivalent to 33.33% [17].

Thus, it is possible to suggest that the *P. heptaphyllum* nanoemulsion can be used as a chemical control for *A. aegypti* at different larval stages at lower concentrations when compared with other plant species.

### 3.4. Adulticidal Evaluation of P. heptaphyllum Nanoemulsion in A. aegypti

This study reported for the first time the evaluation of the adulticidal activity of the *P. heptaphyllum* essential oil nanoemulsion towards *A. aegypti* using the bottle bioassay from the Center for Disease Control and Prevention (CDC).

In another study, the essential oils of *Spathellia excelsa* and *Annona coriacea* did not show adulticidal activity in *A. aegypti* at 0.05 µg·mL^−1^ [18]. The extract of *Annona coriacea* showed a low mortality for the adulticidal activity to *A. aegypti*; the observed mortality at 1000 µg·mL^−1^ was only 6.6% [13]. Researchers performed an adulticidal assay with the nanoformulation of *Heliotropium indicum*, which showed 98.2% mortality to *A. aegypti* mosquitoes at 250 µg·mL^−1^ in 24 h of exposure [16].

Comparing these results above with the concentrations and the time of diagnosis found for the *P. heptaphyllum* essential oil nanoemulsion, it is possible to conclude that our nanoformulation could be used as a chemical control for the *A. aegypti* mosquito in adulthood.

### 3.5. Repellency Evaluation of P. heptaphyllum Nanoemulsion in A. aegypti

The repellent effect of natural products from plants has been identified as an effective way to prevent the proliferation of disease-causing mosquitoes. Repellent action reduces the permanence of mosquitoes in anthropogenic environments, thereby decreasing oviposition and consequently the insect’s life cycle, bringing benefits to the human population [20].

Essential oils are obtained from renewable resources, are rapidly degradable, have a low resistance potential, easy access and a low production cost. As the Amazon region is an area with rich and diverse flora, this type of study has been widely developed and has shown good results [21]. Plants and their derivatives have demonstrated an optimized repellent action of less than 2 h of protection [22].

Assessing the repellent activity, the compound *p*-menthane-3,8-diol in the concentration of 30% was isolated from *Corymbia citriodora* oil, showing satisfactory protection over the test time [23]. In another study that used citronella essential oil in concentrations of 5% and 10%, an expressive repellent action was shown, with average protection rates of 98.1% and 99.0%, respectively [24]. The authors carried out a study with thymol nanocapsules at a concentration of 500 µg·mL^−1^ and pointed out the repellent effect on *A. aegypti* over the 60 and 180 min test, corresponding to a repellency percentage of 67% and 33% in each time [25].

The results indicate that our nanoemulsion formulation has clear potential to be used as a repellent against *A. aegypti*.

## 4. Materials and Methods

### 4.1. Collection of P. heptaphyllum Resin and Botanical Identification

Authorization was given to collect the botanical material from a native forest located at the municipality of Porto Grande (Amapa State, Brazil, lat 0°4′1′′ N lon 51°3′10′′W). A part of the plant that was suitable for botanical identification was collected for exsiccate preparation, which was deposited at the herbarium from the Institute of Scientific Research and Technology from the Amapa State, registered under the code 019059. Moreover, the resin was manually collected from cuts in the *P. heptaphyllum*. Then, the resin was conducted to the laboratory for essential oil extraction.

### 4.2. Extraction of the Essential Oil from the P. heptaphyllum Resin

Then, 100 g of resin was crushed in a porcelain gral and transferred to an Erlenmeyer with 3 L of distilled water. The resin essential oil was extracted with the Clevenger apparatus using the steam dragging technique for 3 h (São Paulo, Brazil) to extract the essential oil through the steam distillation technique. The extraction was carried out in quintuplicate and the oil was stored in an amber glass at 5 °C.

### 4.3. Gas Chromatography Coupled to Mass Spectrometry (GC–MS) Analysis

The phytochemical profile of the essential oil of *P. heptaphyllum* resin was identified by GC–MS using Shimadzu equipment, model CGMS-QP2010 Ultra (Kyoto, Japan), equipped with an RTX-5MS capillary column (30 m × 0.25 mm, film thickness 0.25 μm), with the stationary phase at 5% diphenyl and 95% dimethyl-polysiloxane. The oven temperature was programmed at 60–250 °C at a heating rate of 3 °C/min. The ion source was adjusted to 200 °C and the electronic ionization to 0.70 kV. Helium was the carrier gas at a flow rate of 1.0 mL min^−1^ and an inlet pressure of 57.0 KPa.

A sample of the essential oil was diluted in hexane and 10 µL of that solution was injected into the GC–MS. The relative concentrations (%) corresponding to the essential oil components were calculated using Shimadzu software (Kyoto, Japan). Peak identification was performed by comparing retention indices calculated from an n-alkane series (C_9_ to C_17_) and by the fragmentation pattern of the mass spectrum and compared with data from the equipment library.

### 4.4. Preparation and Characterization of the Nanoemulsion

The nanoemulsion of the essential oil from the resin was prepared by phase inversion without heating, following the titration method [26]. The formulation can be seen with details in our previous publication [10], as well as the characterization through the physical-chemical evaluation of the particle size, polydispersity index and zeta potential (Malvern, UK).

### 4.5. Ovicidal Activity Assay

*A. aegypti* eggs were obtained at the Medical Entomology Laboratory (IEPA) and were kept under standardized climatic conditions in a room (3 m × 4 m) with controlled temperature and humidity (26 ± 2 °C and 80 ± 5%, respectively) and a 12 h photoperiod as recommended by the World Health Organization (WHO) [27].

The most stable *P. heptaphyllum* nanoemulsion (HLB = 14) was used to assess the ovicidal potential. The nanoemulsion (25,000 µg·mL^−1^) was diluted with distilled water in Petri dishes (100 mL) at 2, 5, 10, 15, 20 µg·mL^−1^ and 25 ug·mL^−1^ Each treatment received 25 *A. aegypti* eggs and the percentage of hatching eggs was evaluated after 24 and 48 h. The test was performed in triplicate. The method used to evaluate the effect of the essential oil from the resin of *P. heptaphyllum* was based on the work of Suman et al. [11], where the test concentrations were compared with the results from the positive control (Esbiothrin) and the negative controls sorbitan monooleate and polysorbate 80 (Sigma-aldrich^®^, St. Louis, MO, USA).

### 4.6. Pupicidal Activity Assay

The most stable *P. heptaphyllum* nanoemulsion (25,000 µg·mL^−1^) was diluted with distilled water (100 mL) at 80, 90, 100, 130 and 160 µg·mL^−1^ and 25 *A. aegypti* pupae were transferred to each beaker. The blend of surfactants used in the nanoemulsion (sorbitan monooleato and polysorbate 80) and Esbiothrin were used as positive and negative controls, respectively. The tests were carried out in triplicate and the percentage of pupal mortality was verified after 24 and 48 h of exposure and were considered as dead when they did not present any movement.

### 4.7. Adulticidal Activity Assay

The adulticidal potential of *P. heptaphyllum* nanoemulsion was evaluated using the Center for Disease Control and Prevention’s (CDC) bottle bioassay [28] to determine the diagnostic dose that would kill 100% of female mosquitoes in a diagnostic time.

The nanoemulsion (HLB = 14) was diluted in acetone at concentrations of 100, 150 and 200 µg·mL^−1^, 1 mL of each solution was applied in Wheaton bottles (250 mL) and the test maintained a negative control with the blend of surfactants used in the production of the nanoemulsion.

The solutions were distributed to all internal areas of the bottles by horizontal and vertical rotation movements and were opened at room temperature to evaporate the solvent. Then, 25 *A. aegypti* adult females were transferred to each bottle and mortality was assessed every 15 min for a period of 120 min. Mosquitoes that slid when the bottle was rotated around its axis were considered dead.

### 4.8. Repellent Activity Assay

This study was submitted to the Ethics and Research Committee and received approval through the Certificate of Presentation for Ethical Appreciation (CAAE) number 62616416.0.0000.0003 which ensures the ethical guidelines for research with human beings.

*A. aegypti* adult females of 4 to 15 days old age were obtained from the (IEPA) and 20 volunteers (5 men and 15 women) between 18 and 45 years old participated in the study. Medical history reporting the absence of allergic reactions was used as an inclusion criterion in the research.

This trial employed a standard protocol for the evaluation of repellents in humans recommended by the World Health Organization [29]. *P. heptaphyllum* nanoemulsion (HLB = 14) at the prepared concentration of 25,000 µg·mL^−1^ at concentrations of 100, 150 e 200 µg·mL^−1^ and 1 mL of the solution were applied to the right arm. The negative control was composed of the blend of surfactants used in the nanoemulsion.

The volunteer’s forearms (3 cm × 10 cm area on the right forearm) were inserted into the cage with 50 female mosquitoes with 24 h prior fasting and were exposed for a period of 3 min. Then, the number of mosquito landings for an interval greater than 15 s and the number of bites on the forearms of the volunteers were counted. The test was performed in triplicate for each concentration and the controls for 180 min.

### 4.9. Statistical Analysis

The experimental design used in the ovicidal, pupicidal, adulticidal and repellency trials was randomized and in triplicate. The data were analyzed using mean and standard deviation and analyzed using the Prisma 5.03 program (San Diego, CA, USA). The probabilistic level of error employed was 95%.

## 5. Conclusions

Essential oils are present in plants and have volatile aromatic compounds originating from their secondary metabolism. The nanoemulsion prepared from the essential oil of *P. heptaphyllum* resin is capable of producing innovative products that can combat different development phases of the mosquito *A. aegypti*.

The efficacy of the essential oil in relation to ovicide, pupicide, adulticide and repellency activities was obvious. It is well known that earlier stages of *A. aegypti* are more susceptible. In this study, 100% of pupa mortality was observed at 160 μg·mL^−1^, a concentration lower than that recommended by the literature for alternative potential larvicidal natural products. This follows the concepts of colloids, which may enhance the bioactivity of natural products, including EOs, therefore leading to higher mortality for less susceptible organisms (as would be expected when compared pupa × larva). Therefore, this study contributes to the state of the art of the field and also to the production process of nanoemulsions, which proved to be a sustainable mechanism that values the knowledge of traditional communities in the Amazon and presents itself as an effective nanobiotechnological product in the chemical control of *A. aegypti* that can be used to mitigate important public health epidemics.

## Figures and Tables

**Figure 1 molecules-26-06439-f001:**
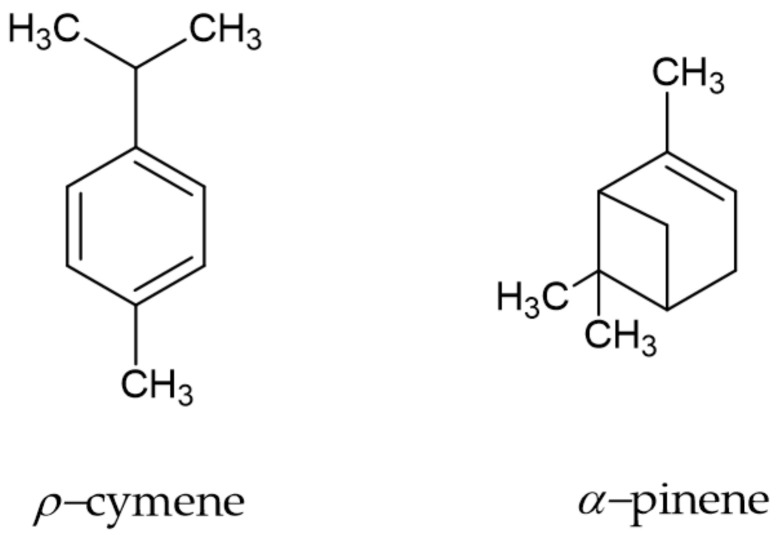
Molecular structure of the major phytochemicals of the *P. heptaphyllum resin* essential oil.

**Figure 2 molecules-26-06439-f002:**
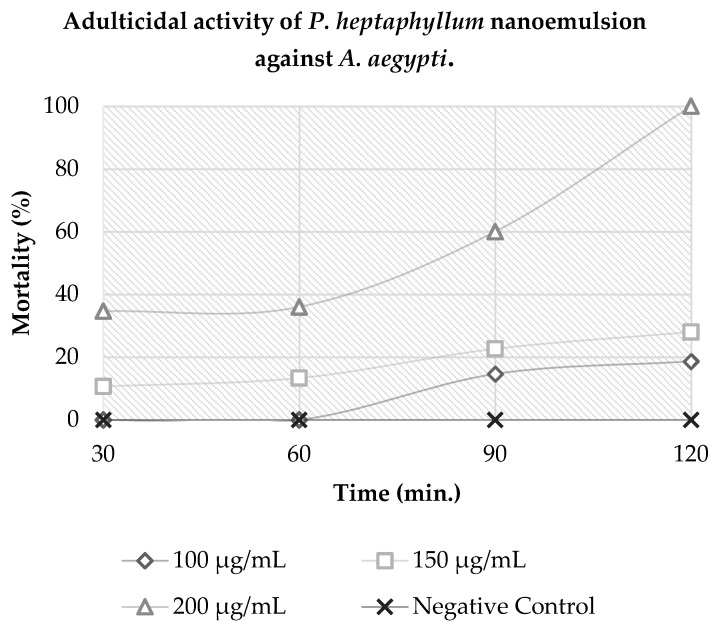
Adulticidal activity of *P. heptaphyllum* nanoemulsion against *A. aegypti*.

**Figure 3 molecules-26-06439-f003:**
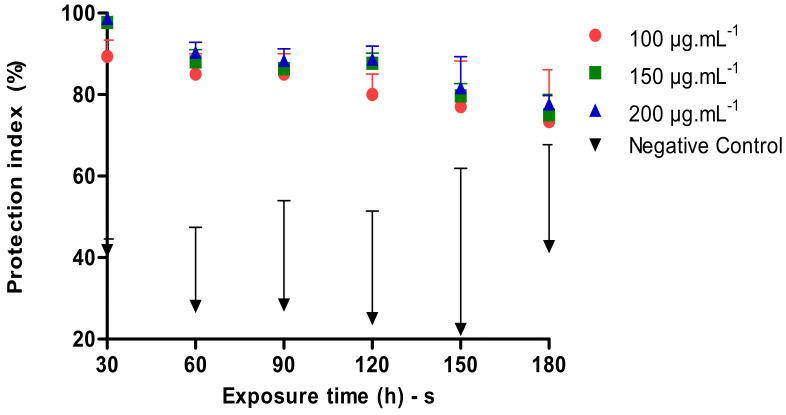
Repellent activity of *P. heptaphyllum* nanoemulsion against *A. aegypti*.

**Table 1 molecules-26-06439-t001:** PHEON Lethal Concentration against *A. aegypti* eggs.

Lethal Concentration	24 h	48 h	*p*-Value
LC_50_ (µg·mL^−1^)	0.404 (µg·mL^−1^)	0.482 (µg·mL^−1^)	0.005
LC_90_ (µg·mL^−1^)	0.060 (µg·mL^−1^)	0.089 (µg·mL^−1^)	0.005
Esbiothrin [11]	0.0034 (µg·mL^−1^)		

**Table 2 molecules-26-06439-t002:** The Lethal Concentration of PHEON against *A. aegypti* pupae.

	24 h	48 h	*p*-Value
LC_50_ (µg·mL^−1^)	99.809 (µg·mL^−1^)	119.036(µg·mL^−1^)	0.005
LC_90_ (µg·mL^−1^)	85.547(µg·mL^−1^)	98.441(µg·mL^−1^)	0.005
Esbiothrin [11]	0.0034 (µg·mL^−1^)		

## Data Availability

Not applicable.

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
