# Peer review of "Biocidal Activity of a Nanoemulsion Containing Essential Oil from Protium heptaphyllum Resin against Aedes aegypti (Diptera: Culicidae)"

_molecules, 2021, doi:10.3390/molecules26216439_

Round 1
Reviewer 1 Report
The paper Biocidal Activity of Nanoemulsion Containing Essential Oil from Protium heptaphyllum Resin Against Aedes aegypti (Dip-tera: Culicidae) is interesting.
However some corrections and explaination can be performed before publication.
I suggest major revision.
Introduction
Currently, the growing concern of society for products that are increasingly safe for the environment and humans, therefore causing lower harmful effects to non-target organisms, has led to a resurgence in the search for plant-derived insecticides [5]. Among natural products for insecticidal purposes, essential oils are considered important candidates for the control of arthropods of medical importance [6].
Please, report other reference to improve the effetct of essential oils not also concernning arthropods of medical importance but also in food prevention.
In this context add referenc such as settings
Open AccessArticle
Flavouring Extra-Virgin Olive Oil with Aromatic and Medicinal Plants Essential Oils Stabilizes Oleic Acid Composition during Photo-Oxidative Stress Agriculture 2021, 11(3), 266.
Figure 1
please improve figure quality
Page 9 material and methods
The electronic ionization to 0.84 kV
Why authors use 0.84 kV and not use 0.70 kV?
Generaly, all librery in GC-MS are obtained using 0.70 kV
Author Response
Point 1: Does the introduction provide sufficient background and include all relevant references?
Response 1: The reference was inserted in the introduction as requested shown in lines 67 to 74.
Point 2: Is the research design appropriate?
Response 2: The design of the experimental research was improved in the following points: methodology, results, discussion and conclusion. To meet the proposed objectives in an understandable way.
Point 3: Are the methods adequately described?
Response 3: The methodology has been improved in the chromatography sections, Collection of P. heptaphyllum resin and botanical identification, Preparation and Characterization of Nanoemulsion.
Point 4: Are the results clearly presented?
Response 4: The authors decided to remove table 1 and Figure 2 to avoid similarity with the previous publication from the author in the same journal. And detail about the resin collection, and nanoemulsion preparation and characterization were provided according to the reviewer's comments 2.
Point 5: Are the conclusions supported by the results?
Response 5: It was improved second paragraph.
Point 6: Figure 1 please improve figure quality
Response 6: The quality of figure 1 has been improved, as requested by the reviewer in lines 97 to 98.
Point 7: Page 9 material and methods The electronic ionization to 0.84 Kv Why authors use 0.84 kV and not use 0.70 kV? Generaly, all librery in GC-MS are obtained using 0.70 kV
Response 7: The operating conditions of the GC-MS were standardized according to the scientific literature, however, a writing error changed from 0.70 to 0.84 kV. In this way, we are correcting the information as requested.

Reviewer 2 Report
Please see attached file

Author Response
Point 1: How was the Protium resin collected? Were trees tapped to collect resin or was naturallyextruded resin collected where available?
Response 1: Resin collection was carried out manually from native plants with a cutting tool and placed in bags, other parts of the plant were collected and photographed to carry out the identification of the species. In the laboratory, the resin was washed to remove dirt. Soon after, its extraction was performed in a clevenger device. The resin was collected naturally, it was not necessary to make inductions in the trees.
Point 2: Why wasn’t larvicidal activity assessed? This is a far more common yardstick than bioassays using other stages.
Response 2: The larvicidal activity has already been performed and published in Molecules doi:10.3390/molecules25225333.
Point 3: Where is the pupicidal data in the Results section?
Response 3: They were already inserted in the template in section 2.4.
Point 4: A positive control is mentioned in the Methods section (Esbiothrin), but the source is not mentioned, but more importantly, no data for this positive control is provided in the Results except in Table 2.
Response 4: The positive control Esbiothrin was referenced from a study to compare with the negative control and other concentrations. Comments were included on the results and methodology.
Point 5: Why wasn’t the essential oil, formulated as a simple emulsion, included in the bioassays for comparison? How do we know that the nanoemulsion is superior?
Response 5: In this study, experimental field conditions did not allow comparing the larvicidal action of the essential oil and its nanoformulation due to the worldwide pandemic situation, however, we intend to include this objective in the next study to contribute to the knowledge regarding the biological action of the nanoemulsion and the essential oil.
Point 6: Why aren’t all the constituents of the essential oil listed in a table? As cymene and pinene make up half of the mass of the oil, why wasn’t a simple binary mixture of these two pure compounds tested as well as the natural oil? It might be more efficient and cost effective to use a mixture of these two compounds rather than producing the essential oil. Presumably the natural resin is of much higher value for medicinal purposes.
Response 6: Chemical constituents were published in a previous article. Thus, minor details of chemical data have been suppressed to avoid repetition. However, they can be found in the following article https://doi.org/10.3390/molecules25225333. We cannot say that the mixture of these two compounds is more efficient, given that the biological activity can be a result of the synergism of all chemical constituents. However, this is a possibility that could be verified in future studies. Natural resin is a raw product obtained directly from collection in the forest, while essential oil undergoes an extraction process, which, depending on the conditions, can represent more cost for the industrial process. Phytochemical knowledge represents the possibility of developing bioproducts oriented by the chemosystematics of the species, not only in its insecticide products, but for other purposes in the pharmaceutical industry.
Point 7: The final sentence of the Discussion reads, “Important to note that the studied nanoemulstion formulation does not lose its effect over time which is an advantage to the development of essential oil-derived products since they are extremely volatile.” One of the reputed advantages of nanoformulations of essential oils is their environmental persistence. Where is the evidence for this important aspect in the present study?
Response 7: Our previous studies (https://doi.org/10.3390/molecules25225333) demonstrated residual larvicidal activity for 11 consecutive days using the same formulation. However, these data were not included in this manuscript, so the authors agree that there is no basis for this statement and deleted from the text, given that their exclusion does not compromise the overall understanding of the manuscript.
Point 8: Line 354: “notorious” is the wrong word to use here. I think the authors meant to say “obvious”
Response 8: The suggestion was made.
Point 9: Review Report Form
Response 9: All points relating to the structure of the text: Introduction, research project, results, discussion, methodology and conclusion. Have been improved. They are highlighted as suggested by reviewers.
Point 10: We noted that there are some parts in your manuscript are similar to the sentences in the published papers. We have used iThenticate to check the plagiarism. The attachment is the iThenticate report. Please carefully check and revise your manuscript for avoiding the duplication. I am writing this letter in the hope that you could modify the data in figure 2, table 1 and the expression in the fourth part of the experimental part. If you need to quote the table or picture directly, please provide a copyright statement
Response 10: The authors decided to remove table 1 and Figure 2 to avoid similarity with the previous publication from the author in the same journal. And detail about the resin collection, and nanoemulsion preparation and characterization were provided according to the reviewer's comments.

Reviewer 3 Report
- The authors should test the ovicidal activity of p-cymene and α-pinene against A. aegypti eggs to compare with that of PHEON .
- The adulticidal activity of p-cymene and α-pinene against A. aegypti mosquitoes also should be test.
Author Response
Point 1: The authors should test the ovicidal activity of p-cymene and α-pinene against A. aegypti eggs to compare with that of PHEON .
Response 1: Dear reviewer, it was not the focus of our study to work with isolated compounds. The intention of the research is to demonstrate the potential of the nanoemulsion. It is unfeasible to carry out a trial with the major compounds in view of the pandemic situation we are experiencing, it remains for future work.
Point 2: The adulticidal activity of p-cymene and α-pinene against A. aegypti mosquitoes also should be test.
Response 2: Dear reviewer, it was not the focus of our study to work with isolated compounds. The intention of the research is to demonstrate the potential of the nanoemulsion. It is unfeasible to carry out a trial with the major compounds in view of the pandemic situation we are experiencing, it remains for future work.

Round 2
Reviewer 1 Report
All correctuons were made. Paper can be accept
Author Response
there were no changes.
Reviewer 2 Report
The revised manuscript is much improved over the original version. Regarding the positive control, the authors should provide the source of this material (in the Methods section), as it was used in the present study, rather than providing a reference to use of this material in a previous paper.
Author Response
Point 1: The revised manuscript is much improved over the original version.
Regarding the positive control, the authors should provide the source of this
material (in the Methods section), as it was used in the present study, rather
than providing a reference to use of this material in a previous paper.
The method used to evaluate the effect of nanoemulsion the
essential oil from the resin of Protium heptaphyllum was based in the
work of Suman et al. [29], where the test concentrations were compared
with the results from the positive control (Esbiothrin) and the negative
controls (sorbitan monooleate and polysorbate 80).
Reviewer 3 Report
The manuscript can be accepted in present form.
Author Response
there were no changes.